# How Do Lactobacilli Search and Find the Vagina?

**DOI:** 10.3390/microorganisms11010148

**Published:** 2023-01-06

**Authors:** Gregor Reid

**Affiliations:** 1Departments of Microbiology and Immunology, and Surgery, University of Western Ontario, London, ON N6A 3K7, Canada; gregor@uwo.ca; 2Lawson Health Research Institute, London, ON N6A 4V2, Canada

**Keywords:** *Lactobacillus* species, vagina, microbiome, evolution, mechanisms of persistence

## Abstract

With the advent of omics technology and the improved culturing of anaerobic microbes, there is a good understanding of the microorganisms present in a healthy and diseased vagina. This has led to the identification of a select few *Lactobacillus* species associated with health. However, the origin of these species and how they reach the vagina remains unclear, as does their timing of colonization. In an effort to create badly needed therapies for women, these gaps in knowledge need to be addressed.

## 1. Introduction

In this Special Issue’s information page, which is entitled “Vaginal Microbiota: Impact on Health and Disease” (https://www.mdpi.com/journal/microorganisms/special_issues/vaginal_microbiota_health_disease; accessed on 29 December 2022), it is stated that “The vaginal microbial community is characterized by the presence of numerous microorganisms in a dynamic balance between themselves and the host”. We know this to be true based on old culture systems and modern omics technologies. However, what we do not know is how this community evolved to be so critical for health and reproduction.

Microbiota studies can document the stages of vaginal colonization from birth to reproductive age. The result is not universal, but in most cases, lactobacilli form the backbone of the microbiota in healthy women. The big question is: out of over 200 species, how did only a few *Lactobacillus* species search for and find the vaginal niche and then become the dominant colonizer? The scientific answers tend to range from vague to circumstantial. However, really, we do not know how they get there and stay.

## 2. Are There Answers?

A recent study of 151 vaginally delivered and formula-fed newborn infants showed that lactobacilli were present in the gut in samples taken between days 21 and 30 [1]. The species detected were *Lacticaseibacillus rhamnosus*, *Lacticaseibacillus paracasei*, *Limosilactobacillus fermentum*, *Lactobacillus delbreuckii*, *Lacticaseibacillus casei* and *Lactobacillus acidophilus*. This is interesting and raises several additional questions.

Why do they not become the dominant species in the urogenital tract through an ascension from the rectum? This is the mechanism that is currently viewed as making the most sense. Additionally, it is believed that the first bacterial species that inhabit a site, stay there and make it difficult for exogenous strains to latterly colonize. Indeed, that is one explanation for probiotic lactobacilli not being able to persist when implanted into the vagina.

The main species found to dominate the vagina later in life comprise *Lactobacillus iners*, *Lactobacillus crispatus*, *Lactobacillus jensenii* and *Lactobacillus gasseri*, which I will refer to as “the Key4” [2] with *Limosilactobacillus vaginalis Limosilactobacillus fermentum, Lactobacillus helveticus* or other species in low abundance [3]. How do they reach the vagina and dominate? The ability to produce lactic acid per se cannot be a primary reason as all these species do this, albeit the amounts and anti-inflammatory effects may be important [4] along with improving epithelial barrier function [5]. Arguably the isomer form does not seem to be critical given that *L*. *crispatus* and *L. gasseri* produce both d- and l-lactic acid, *L. iners* the l-isomer and *L. jensenii* the d-isomer [6].

The origin of species does not stand out as being important since lactobacilli are found in a variety of other animals and food substances, particularly dairy [7]. An excellent review has explored some of these issues and assigned lactobacilli into three main lifestyle categories: free living, host-adapted and nomadic [7]. The authors found that these were associated with phylogenetic grouping. This adaptation concept is further supported by genomic analysis of *L. iners* showing traits that would allow it to persist [8], including its ability to adhere to human fibronectin [9]. It is further supported by genomic and phenotypic analyses of three strains of *L. rhamnosus*, which show that the GR-1 vaginal probiotic had a unique cluster for exopolysaccharide production, potentially making it more suitable for that habitat than the GG and LC705 strains [10] Another study we performed showed that within the *L. crispatus* species there are metabolites supportive of vaginal health [11]. The problem is that none of this evidence indicates how these strains reached the vagina and then out-competed other microorganisms to allow them to persist.

The succession of species colonizing the oral cavity provides an example of a process that appears to be well understood, albeit in general terms [12]. However, there are no studies indicating this occurs in the vagina, whereby some species colonize and ‘prepare’ an environment for others to eventually take over. If such a succession was to occur, it might happen at puberty and be influenced by hormones. However, there are no data to show that only the growth or adherence of the Key4 is enhanced by hormones, including estrogen. Furthermore, the ability to adapt to the vaginal pH or to produce substances that inhibit uropathogens are not solely properties of these four species. How the Key4 or, indeed, other *Lactobacillus* sp. were present in the vagina during puberty is still not resolved.

When thinking of evolution, what factors could have led to the selection of bacterial species for vaginal habitation? The ability to adapt to seminal fluid seems an unlikely trait, given that the species are found in virgins and lesbians [13,14]. Is it possible that the Key4 improve the chance of sperm fertilizing the egg? One clinical study explored this topic and showed that implantation was more successful when a *Lactobacillus*-dominated microbiota was present, but they did not examine this at the species level [15]. If there was a correlation with the Key4, what would be the mechanism, and how is it more expressed in these species than in others? The authors suggested that lactic acid and short-chain fatty acid production were not involved, but anti-inflammatory mediators might play a role. Many studies have examined the anti-inflammatory effects of lactobacilli, including those of *L. crispatus*, so if this is a potential mechanism, further targeted studies are necessary.

In terms of host factors being preferential to different *Lactobacillus* species, there is no evidence of this in mucus or epithelial cell surfaces, although the lowered ability to degrade protective mucus may be a feature in favor of *L. crispatus* [16]. A more significant nuance is the unique ability of some strains of *L. crispatus* to utilize glycogen [17].

Another consideration could be the metabolic efficiency of certain species within the milieu of the vagina. The idea is that some species better utilize the nutrients available in the vagina than other strains. A study of two strains has shown differences [18] but interpreting these across species would require significantly more research.

In summary, it cannot be by chance that so few species dominate the vaginal canal in healthy women around the world who consume very different diets, yet no traits have been identified explaining this property. This would seem to be a critical knowledge gap that needs investigation before we can truly understand the vaginal microbial community, how it evolves and adapts, and how to explain the impact of disturbances.

## 3. What’s Next?

One way to possibly determine how and when one of the Key4 species becomes present and dominant is to perform sequential culture and sequencing studies plus record dietary intake and test samples for the species. The starting point would be when none of these species are identified in the vagina prior to puberty. This represents an enormous and tedious experiment requiring vaginal and stool sampling with onerous compliance from the test subjects and hormonal testing to verify puberty onset.

Supposing such an experiment verified that this was indeed the time that *L. crispatus*, for example, entered the niche and it was identified in chicken, or a fermented food consumed by the subject. Then, a study of the strain showed it had properties highly associated with the maintenance of a healthy vagina. So what? Would it lead to recommendations that girls entering puberty should consume certain food products to establish a health-promoting vaginal microbiota? Would it lead to isolating the strain and determining if it could be a probiotic able to colonize other young girls, and if so, what are the ethical implications?

If a second subject acquired *L. jensenii* and a third became colonized with *L. iners*, how would this alter the recommendations?

Would any of these freshly isolated strains be able to persist in the vagina of post-menopausal women afflicted with recurrent infection, or would estrogen be required as an adjunctive therapy?

I realize that this article raises far more questions than it answers, but that is the basis from which scientific endeavor emerges. The collective view may be that it does not matter how and when certain species become dominant in the vagina, as there is little that can be done to influence it. I would counter by saying that we need to understand what we do not know, especially for health maintenance. Rightly, studies up to this point have focused on infection and disease, but mostly these have failed to create new therapies that lower the incidence or prevent recurrences. Going forward, we need to utilize new technologies, perhaps the vagina-on-a-chip model [19], to determine the factors responsible for resilient health-promoting microorganisms to persist in the vagina. Then, see if these factors can be enhanced to allow probiotics or the women’s own strains to be administered when dysbiosis or chronic disease occurs.

As of today, since no probiotic strains, including members of the Key4 or strains chosen for specific anti-pathogen properties [20,21], can persist in the vagina regardless of their mode of application, they must be continually administered. This illustrates that we have still not mastered strain selection, their ideal phenotype as they enter the vagina, or the necessary properties for them to adapt to, and persist in, this environment. 

## Data Availability

Not applicable.

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
