# Peer review of "How Do Lactobacilli Search and Find the Vagina?"

_microorganisms, 2023, doi:10.3390/microorganisms11010148_

Round 1

Reviewer 1 Report

This paper addresses a question of a large scientific importance, i.e., the origin of vaginal lactobacilli. Indeed, there is a knowlegde gap in our understanding how and when few Lactobacillus species (L. iners, L. crispatus, L. jensenii, and L gasseri) became the dominant colonizers of the vagina. The paper outlines questions to be answered to fill this gap. In general, this is an interesting opition paper. The only remark I would like to make is about the author's concept of Key5, which includes Lactobacillus vaginalis (or Limosilactobacillus vaginalis) in the key Lactobacillus species. This species, although not so rare in vaginal samples, is present in extremely low proportions (commonly far below 1%), so it appears unlikely that it exerts biological effects on its own. 

Author Response

The reviewer makes a good point and it could be argued that a few different species could be the fifth most common depending on the population. I have added this point, plus about low abundance to the text.

Reviewer 2 Report

I have read with great interest this manuscript.

First of all, I would like to suggest the author to revise the refernces, in order to update the list.

Moreover, I would add a paragraph about the importance of lactic acid L and D isomers, since they are differently produced by different species of Lactobacilli.

Finally, I appreciate that the author himself states that this article raises far more questions than answers, but I think that he should make a position on the use of vaginal probiotics, especially on the use of vaginal strains of lactobacilli

Author Response

The reference list was to cover the expanse of time to illustrate that despite research important questions remain. Nevertheless, I have now recent added papers from 2020 (ref 4), 2021 (ref 3), 2022 (ref 5) and expanded discussion of lactic acid isomers.

I added a comment at the end on probiotic application.

Reviewer 3 Report

Manuscript ID Microorganisms-2133371

Reviewer report

Congratulations to Dr. Gregor Reid, his Opinion Article on how certain lactobacilli (described by the author as “the Key5”) are found in the vaginal microbiota. The present work is very interesting for the scientific community worldwide and deserves to be published without any doubt. 

The manuscript is well-written and should be published. I recommended some improvements that should beneficiate the manuscript. However, it is important to mention that none of the comments or recommendations are mandatory and only should be followed if the author considers them useful for his Opinion Article.

All is summarized in my minor comments. Please check below.

Minor comments

Lines 10 and - Please replace “organisms” with “microorganisms”. Also, replace the same term in the remaining manuscript.

Line 15- Please put “Lactobacillus” in italics form. Also, I recommend replacing the keyword “vagina” with “vaginal microbiota” or “vaginal microbiome”, but it is not mandatory.

Line 18- Please cite the special issue to the Readers “In this special issue’s information page, …”, which is entitled “Vaginal Microbiota: Impact on Health and Disease at https://www.mdpi.com/journal/microorganisms/special_issues/vaginal_microbiota_health_disease

Lines 58-64- “Yet, there are no studies indicating this occurs in the vagina whereby some species colonize and ‘prepare’ an environment for others to eventually take over. If such a succession was to occur, it might happen at puberty and be influenced by hormones. However, there are no data to show that only the growth or adherence of ‘the Key5′ is enhanced by hormones, including estrogen. Furthermore, the ability to adapt to the vaginal pH or to produce substances that inhibit uropathogens are not solely properties of these species. Plus, how ‘the Key5′ got to the vagina at puberty is still not resolved.”

It is my knowledge that the Key5 and other lactobacilli are not only acquired at puberty age but start from the time that female is born, like the intestinal or skin microbiota, although it suffers its changes and evolution until adult age. Please also address this point to avoid misunderstanding from the Readers.

Lines 67- Please replace “vaginal habitation” with “vaginal colonization”.

Lines 68-76 “Is it possible that ‘the Key5′ improve the chance of sperm fertilizing the egg? One clinical study explored this topic and showed that implantation was more successful when a Lactobacillus‐dominated microbiota was present, but they did not examine this to the species level [11]. If there was a correlation with ‘the Key5′, what would be the mechanism and how is it more expressed in these species than in others? The authors suggested that lactic acid and short‐chain fatty acid production were not involved but anti‐inflammatory mediators might play a role. However, many studies have examined anti‐inflammatory effects of lactobacilli including for L. crispatus, so if this is a potential mechanism, further targeted studies are necessary.”

I agree with Dr. Reid, and it was already reported the anti-inflammatory effect of certain lactobacilli, such as L. crispatus. In addition, many studies are showing the agglutination or immobilization of sperm and therefore selecting the strongest ones for egg fertilization, I send you some examples of these studies:

https://www.frontiersin.org/articles/10.3389/fcell.2021.705690/full 

https://www.frontiersin.org/articles/10.3389/fcimb.2020.620529/full

I do believe that it is worth mentioning as other lactobacilli function as part of the vaginal microbiota if the author considers it relevant (it is not mandatory).

Lines 81-84- “Another consideration could be metabolic efficiency of certain species within the mileau of the vagina. The idea is that some species better utilize nutrients available in the vagina than other strains. A study of two strains has shown differences [14] but interpreting these across species would require significantly more research.”

I agree with Dr. Reid that metabolic efficiency and the interaction of lactobacilli with environmental factors (such as nutrients) is an important variable to be considered by the Readers. Also, there is evidence that clusters are formed by certain Lactobacillus species acting as a probiotic consortium, leading to the consideration that the probiotic effect of lactobacilli should also be analyzed as a total group instead of a single Lactobacillus species by itself. One example of this idea is reported in the following publication:

https://www.frontiersin.org/articles/10.3389/fcimb.2022.863208/full

Likewise, I do believe that it is worth mentioning that the probiotic activity is caused not only by individual Lactobacillus species but also by its multi-microbial interaction. However, the probiotic activity promoted by multi-microbial consortia is still unknown. This recommendation is not mandatory, but I do believe that it will improve the author’s point of view if the author considers it relevant.

Line 110-11- “I realise this article raises far more questions than it answers, but that’s the basis from which scientific endeavor emerges.”

It is an important message that must be reinforced for the scientific community.

Again, congratulations to Dr. Gregor Reid for the present Opinion Article. It was an honor to read his opinion on the present matter.

Author Response

Lines 10 - “microorganisms” replaced throughout.

Line 15- “Lactobacillus” is now italicized and microbiome added.

Line 18- “Vaginal Microbiota: Impact on Health and Disease at https://www.mdpi.com/journal/microorganisms/special_issues/vaginal_microbiota_health_disease is now added.

Lines 58-64- It is my knowledge that the Key5 and other lactobacilli are not only acquired at puberty age but start from the time that female is born, like the intestinal or skin microbiota, although it suffers its changes and evolution until adult age. Please also address this point to avoid misunderstanding from the Readers.

The point being made here is we do not know when the species colonize and become dominant or where they come from. It could be in the uterus, upon birthing, at puberty, from the gut or skin, we simply don't know. Given the apparent importance of these species in health, we should be trying to find out the source and when they 'arrive' in the vagina and then become dominant, otherwise we will continue to leave it to chance and perhaps never solve issues of dysbiosis.

Of note, I changed it to 'key4' in response to another reviewer pointing out that any fifth species is in low abundance.

Lines 67- Please replace “vaginal habitation” with “vaginal colonization”.

I prefer to use the word habitation as it refers to "the state or process of living in a particular place" which is what these four main species are able to do.

I appreciate the two papers on fertilization but one was in rats and the other a review which was interesting but the concept that lactobacilli inhibit fertilization would require definitive clinical proof before I would include these references. Moreover, such discussion is too distant from the main focus of the opinion piece.

I do mention other lactobacilli being functional in the vagina, but the question is how did they get there?

Thank you for pointing out the Ecuadorian study which has now been added as ref 21.

Your kind comments are appreciated.

Round 2

Reviewer 2 Report

Many thanks to the author.

I think that the paper can be published in this revised form